# HDAC Screening Identifies the HDAC Class I Inhibitor Romidepsin as a Promising Epigenetic Drug for Biliary Tract Cancer

**DOI:** 10.3390/cancers13153862

**Published:** 2021-07-31

**Authors:** Christian Mayr, Tobias Kiesslich, Sara Erber, Dino Bekric, Heidemarie Dobias, Marlena Beyreis, Markus Ritter, Tarkan Jäger, Bettina Neumayer, Paul Winkelmann, Eckhard Klieser, Daniel Neureiter

**Affiliations:** 1Center for Physiology, Pathophysiology and Biophysics-Salzburg and Nuremberg, Institute for Physiology and Pathophysiology-Salzburg, Paracelsus Medical University, Strubergasse 22, 5020 Salzburg, Austria; tobias.kiesslich@pmu.ac.at (T.K.); sara.erber@outlook.com (S.E.); dino.bekric@pmu.ac.at (D.B.); heidi.dobias@pmu.ac.at (H.D.); marlena.beyreis@pmu.ac.at (M.B.); markus.ritter@pmu.ac.at (M.R.); 2Department of Internal Medicine I, University Clinics Salzburg, Paracelsus Medical University, Müllner Hauptstrasse 48, 5020 Salzburg, Austria; 3Ludwig Boltzmann Institute for Arthritis und Rehabilitation, Paracelsus Medical University, Strubergasse 22, 5020 Salzburg, Austria; 4School of Medical Sciences, Kathmandu University, Kavreplanchowk, Dhulikhel 45200, Nepal; 5Department of Surgery, University Clinics Salzburg, Paracelsus Medical University, Müllner Hauptstrasse 48, 5020 Salzburg, Austria; ta.jaeger@salk.at; 6Institute of Pathology, University Clinics Salzburg, Paracelsus Medical University, Müllner Hauptstrasse 48, 5020 Salzburg, Austria; b.neumayer@salk.at (B.N.); paul.winkelmann@stud.pmu.ac.at (P.W.); e.klieser@salk.at (E.K.); d.neureiter@salk.at (D.N.); 7Cancer Cluster Salzburg, 5020 Salzburg, Austria

**Keywords:** histone deacetylase inhibitor, biliary tract cancer, romidepsin, HDAC 1, HDAC 2, HDAC class I

## Abstract

**Simple Summary:**

Biliary tract cancer (BTC) is a rare disease with dismal outcomes. Therefore, the investigation of new therapeutic targets is urgently required. In this study, we demonstrate that histone deacetylases (HDACs) are expressed in BTC cell lines and that treatment of BTC cells with different HDAC class inhibitors reduces cell viability. Specifically, we found that BTC cells are vulnerable to the HDAC class I inhibitor romidepsin. Treatment with romidepsin resulted in apoptotic cell death of BTC cells and reduced HDAC activity. Furthermore, romidepsin augmented the cytotoxic effect of the standard chemotherapeutic cisplatin. HDAC class I proteins were also expressed in BTC patient samples. We detected that BTC patients with high HDAC-2-expressing tumors showed a significantly shorter survival. In summary, we were able to demonstrate that BTC cells are vulnerable to HDAC inhibition and that the HDAC class I inhibitor romidepsin might be a promising anti-BTC substance.

**Abstract:**

Inhibition of histone deacetylases (HDACs) is a promising anti-cancer approach. For biliary tract cancer (BTC), only limited therapeutic options are currently available. Therefore, we performed a comprehensive investigation of HDAC expression and pharmacological HDAC inhibition into a panel of eight established BTC cell lines. The screening results indicate a heterogeneous expression of HDACs across the studied cell lines. We next tested the effect of six established HDAC inhibitors (HDACi) covering pan- and class-specific HDACis on cell viability of BTC cells and found that the effect (i) is dose- and cell-line-dependent, (ii) does not correlate with HDAC isoform expression, and (iii) is most pronounced for romidepsin (a class I HDACi), showing the highest reduction in cell viability with IC_50_ values in the low-nM range. Further analyses demonstrated that romidepsin induces apoptosis in BTC cells, reduces HDAC activity, and increases acetylation of histone 3 lysine 9 (H3K9Ac). Similar to BTC cell lines, HDAC 1/2 proteins were heterogeneously expressed in a cohort of resected BTC specimens (*n* = 78), and their expression increased with tumor grading. The survival of BTC patients with high HDAC-2-expressing tumors was significantly shorter. In conclusion, HDAC class I inhibition in BTC cells by romidepsin is highly effective in vitro and encourages further in vivo evaluation in BTC. In situ assessment of HDAC 2 expression in BTC specimens indicates its importance for oncogenesis and/or progression of BTC as well as for the prognosis of BTC patients.

## 1. Introduction

The term biliary tract cancer (BTC) describes a heterogeneous group of malignant tumors arising at different locations within the biliary tract system. [1]. Common risk factors for the development of BTC include chronic hepatitis B and C, cholelithiasis, liver fluke infestation, type 2 diabetes mellitus, obesity, alcohol consumption, and smoking [2]. However, it is believed that most BTC cases are sporadic and not causal to any of the mentioned risk factors [2]. Due to the anatomic and histological heterogeneity of BTC, it is difficult to define common driver mutations. Frequently observed mutations in BTC include *KRAS*, *SMAD4*, and *TP53*, as well as the newly detected and targetable *IDH1/2* and *FGFR2* fusions [2]. Nevertheless, the prognosis for BTC is dismal. The standard chemotherapeutic regimen for patients with advanced BTC comprises a combination of cisplatin and gemcitabine and leads to a median survival of about one year [3]. Consequently, an investigation into alternative therapeutic strategies for BTC is an urgent necessity. 

The term epigenetics describes heritable changes in gene expression that are independent of the primary DNA sequence. Post-translational modifications of histones represent one major mechanism of epigenetic gene regulation and are executed by various specific enzymes. Histone deacetylases (HDACs) are a group of enzymes that deacetylate lysine residues of histones, thereby generally causing transcriptional silencing of genes [4,5]. HDACs can be grouped into four classes. HDAC class I comprises HDACs 1, 2, 3, and 8, which are found in the nucleus [6]. Members of HDAC class IIa (HDACs 4, 5, 7, and 9) and class IIb (HDACs 6 and 10) are found in the nucleus and the cytoplasm [6]. HDAC class IV only comprises HDAC 11 [6]. Mechanistically, these classes are zinc-dependent and referred to as ‘classical HDACs’ [6,7]. In contrast, class III HDACs (sirtuins 1–7) are NAD^+^-dependent [8]. The biological role of HDACs is to generate and maintain the balance between protein (histone) acetylation and deacetylation. Mariadason et al. [9] stated that up to 10% of all genes are epigenetically regulated by HDACs, underlining the importance of HDACs for correct cellular function. Of note, although the term histone deacetylases implies histones as the major targets of HDACs, numerous non-histone targets are known (e.g., p53), which is in line with the cytoplasmic localization of some HDACs [10,11]. 

Aberrantly expressed and/or active epigenetic regulators are crucially involved in the development and progression of cancer. Numerous studies describe a clear connection between HDAC overexpression and tumorigenesis, especially for HDAC class I enzymes [4,12,13]. High expression of HDAC class I is associated with poor prognosis in various tumor types including ovarian cancer, prostate cancer, bladder cancer, breast cancer, and colorectal cancer, as reviewed by Li et al. [12]. Moreover, HDAC class I enzymes are involved in DNA damage response, metastasis (by directly inhibiting the epithelial marker *E-Cadherin*), cell cycle progression, apoptosis resistance, and angiogenesis [6]. 

Consequently, HDACs were identified as attractive therapeutic targets, leading to the development of HDAC inhibitors (HDACis). HDACis can be classified as non-selective ‘pan-inhibitors’ that target all HDACs (e.g., vorinostat and panobinostat), as well as selective inhibitors that specifically target one HDAC class or individual HDACs [6]. The treatment of tumor cells with HDACis results in diverse anti-tumor effects including cell cycle arrest, apoptosis, inhibition of angiogenesis, and alterations in non-coding RNA expression [12]. Moreover, the combination of HDACis with other epigenetic inhibitors or DNA-damaging substances led to promising synergistic effects [6,14,15]. The importance of HDAC inhibition as a viable anti-tumor strategy is underlined by the fact that currently four HDACis are approved by the U.S. Food and Drug Administration (FDA) for the treatment of cancer [6]. In 2006, the pan-inhibitor vorinostat was approved by the FDA for the treatment of progressive and persistent cutaneous T-cell lymphoma. In the following years, two other pan-inhibitors, termed belinostat and Panobinostat, were approved by the FDA for treatment of peripheral T-cell lymphoma and multiple myeloma, respectively [6]. Romidepsin (FK228) is the only selective HDAC inhibitor that is currently approved by the FDA and the European Medicines Agency (EMA) for cancer treatment (cutaneous T-cell lymphoma) [16,17]. Romidepsin selectively inhibits HDAC class I with weak inhibitory activity against class II [17,18]. Interestingly, native romidepsin itself possesses no structural element that is able to interact with the active site of HDACs that would explain its inhibitory effect on HDACs [19]. In this context, Furumai et al. showed that upon cellular uptake, romidepsin is reduced and thereby converted into its active form, which can interact with zinc in the active site pocket of HDAC enzymes [17]. 

Data regarding the involvement of HDACs in BTC are sparse and therefore warrant further investigation. Morine et al. demonstrated that HDAC 1 is overexpressed in BTC and that HDAC 1 expression correlates with higher tumor stages and bad prognosis [20]. Likewise, expression of HDACs 2 and 3 was shown in BTC tumor tissue and associated with poor clinical characteristics and shorter survival [21,22,23]. Substance- or RNA-interference-based inhibition of HDACs in BTC cells resulted in cell cycle arrest, apoptosis, and reduction of cellular invasiveness [24,25,26]. There is also evidence that HDACs are involved in chemoresistance of BTC cells and that treatment with HDACis increases the efficiency of standard chemotherapeutics [27,28,29]. 

Based on these promising data, we here aimed for a broad investigation of HDACis as a potential anti-BTC strategy. Using a comprehensive human BTC in vitro model, we analyzed the expression of the different HDACs as well as the effects of a comprehensive panel of HDACis targeting different HDAC classes on BTC cell viability and demonstrated for the first time the potential of the clinically relevant HDAC class I inhibitor romidepsin in BTC cells. Furthermore, we evaluated immunohistochemically the in situ protein expression of HDAC 1/2 protein in the BTC specimen and its clinical relevance to clinicopathological parameters.

## 2. Materials and Methods

### 2.1. Cell Culture and Substances

A panel of eight BTC cell lines was used in this study: CCC-5 (ACC 810 [30]), EGI-1 (ACC 385 [31]), and TFK-1 (ACC 344 [32]) cells were purchased from the German Collection of Microorganisms and Cell Cultures (DSZM; https://www.dsmz.de, Braunschweig, Germany); HuCCT1 (JCRB0425 [33]), KKU-055 (JCRB1551), NOZ (JCRB1033 [34]), OCUG-1 (JCRB0191 [35]), and OZ (JCRB1032 [36]) were purchased from the Japanese Collection of Research Bioresources Cell Bank (JCRB, Osaka, Japan). The cholangiocyte cell line MMNK-1 was used to evaluate HDAC expression in non-tumor cells (JCRB1554 [37]). Cells were cultured under standard cell incubator conditions (37 °C in a humidified atmosphere containing 5% CO_2_. All cells were cultured in high-glucose Dulbecco’s modified Eagle’s medium (DMEM, Gibco, ThermoFisher Scientific, Waltham, MA, USA) supplemented with 10% (*v*/*v*) fetal bovine serum (FBS, Biochrom, Berlin, Germany), 1% antibiotic–antimycotic (ABAM, Merck, Darmstadt, Germany), 1 mM sodium pyruvate (Pan Biotech, Aidenbach, Germany), and 10 mM HEPES (Pan Biotech).

HDAC inhibitors belinostat, LMK-235, mocetinostat, romidepsin, tubastatin A, and vorinostat were purchased from Selleckchem (Houston, TX, USA) and dissolved in 100% dimethyl sulfoxide (DMSO) into a stock concentration of 20 mM, according to the manufacturer’s protocol. Stocks were aliquoted and stored at −20 °C.

Cells were seeded using the following seeding numbers in transparent 96-well microplates: CCC-5, HuCCT1, KKU-055, NOZ, OCUG-1, and OZ were seeded using 10,000 cells per well; EGI-1 and TFK-1 cells were seeded using 15,000 cells per well. In 60 mm dishes, a total of 9.73 × 10^5^ cells were seeded for CCC-5, HuCCT1, KKU-055, MMNK-1, NOZ, OCUG-1, and OZ, and a total of 1.3 × 10^6^ cells were seeded for EGI-1 and TFK-1.

### 2.2. HDAC mRNA Expression Analysis

Cells were seeded in 60 mm dishes and cultivated overnight, and total RNA was isolated with TRI Reagent^®^ (Merck) and a Direct-zol RNA Miniprep kit (Zymo Research, Irvine, CA, USA). cDNA synthesis was performed with 1 µg of isolated RNA using a GoScript™ Reverse Transcriptase kit (Promega, Fitchburg, WI, USA). Quantitative real-time PCR was performed on a ViiA7 real-time PCR system (Applied Biosystems, ThermoFisher Scientific) using GoTaq^®^ Master Mix (SYBR^®^ Green, Promega). mRNA expression levels of HDACs are presented as ΔCt values related to beta-actin. Primers were purchased as 100 µM stocks (KiCqStart^®^ SYBR^®^ Green Primers, Merck). Primer sequences are listed in Appendix A.

### 2.3. Cytotoxicity of HDAC Inhibitors

For investigation of dose-dependent effects of all HDACis on BTC cell viability, cells were seeded in 96-well microplates using the above-mentioned seeding numbers. After 24 h, cells were washed with a serum-free medium (sfDMEM) and incubated with HDACis for 72 h using a 10-step 1:2 dilution series. Concentration ranges were chosen based on the literature and the manufacturers’ instructions. For belinostat, LMK-235, mocetinostat, and vorinostat, cells were incubated with a 10-step 1:2 dilution series starting at 20 µM (dilution series range: 20–0.04 µM); for romidepsin, cells were incubated with a 10-step 1:2 dilution series starting at 1 µM (1–0.002 µM); for tubastatin A, cells were incubated with a 10-step 1:2 dilution series starting at 100 µM (100–0.2 µM). Incubation of cells with HDACis was performed in sfDMEM to avoid interactions between the substances and serum components.

After 72 h incubation time, viability was measured using resazurin (Alfa Aesar, Haverhill, MA, USA) as described ([38]) on a Spark multimode reader (Tecan, Grödig, Austria). Viability was related to untreated control (UTC) cells. To exclude that changes in the fluorescence intensity following HDACi treatment are due to the autofluorescence effects of the respective HDACis, we measured fluorescence intensity of different concentrations of each HDACi (including the highest concentration used in dilution series) in serum-free media compared to serum-free media only (blanks) and untreated cells (Appendix A). IC_50_ values (concentration at which cell viability was reduced to 50%) were calculated using four-parameter logistic regression.

Time-resolved analysis of the effect of romidepsin on cell viability in KKU-055 and TFK-1 cells was performed using the same seeding numbers as the experiments comprising all HDACis. Viability of romidepsin was measured after 2, 24, 30, and 48 h, respectively, using resazurin and a Spark multimode reader. Viability was related to the corresponding untreated controls. Incubation of cells with romidepsin was performed in sfDMEM to avoid interactions between the substances and serum components.

The mode of cytotoxicity of romidepsin was investigated using the RealTime-Glo Annexin V Apoptosis and Necrosis Assay (Promega) according to the manufacturers’ instructions. In short, cells were seeded in a white-walled 96-well microplate, allowed to grow overnight, and incubated with 5 nM romidepsin. Using the environmental control of the multimode reader (5% CO_2_, 37 °C), luminescence and fluorescence were measured every 30 min for 48 h. A humidity cassette was used to avoid evaporation.

### 2.4. HDAC 1/2 Activity

Baseline HDAC 1/2 activity of BTC cells, as well as the effect of romidepsin on HDAC 1/2 activity, was evaluated using an HDAC-Glo I/II Assay and Screening System (Promega). The assay uses a substrate that is optimized to specifically detect HDAC class I and II activity with a sensitivity about 2.5-fold higher for HDAC 2 than for HDAC 1 and about >20-fold higher than for the other HDAC isoforms. Therefore, we used this assay to evaluate HDAC 1 and 2 activity. For measurement of baseline HDAC 1/2 activity, cells were seeded in a 96-well microplate, allowed to grow overnight, and assayed according to the manufacturer’s instructions using a Spark multimode reader (Tecan). For measurement of the effect of romidepsin on HDAC 1/2 activity, cells were seeded in a 96-well microplate, allowed to grow overnight, and incubated for 24 h with 5 nM romidepsin before measuring the HDAC 1/2 activity.

### 2.5. Western Blot

Cells were seeded in 60 mm dishes, allowed to grow overnight, and incubated with 5 nM romidepsin. After 24 h, cells were harvested, centrifuged, counted, and frozen as pellets at −20 °C until further use. Frozen pellets were thawed and resuspended in a defined volume phosphate-buffered saline (PBS) to achieve a concentration of 10^7^ cells per ml. Cells were lysed by sonification (10 pulses, Sonopuls HD70, UW 70 ultrasound head, Bandelin) and centrifuged. A total of 10 µL of the supernatant was mixed with 10 µL 2x sodium dodecyl sulfate (SDS)-containing a lysis buffer, incubated for 5 min at 95 °C and centrifuged to collect the condensate. Samples were run on gradient SDS gels (4–20% Mini-PROTEAN gels) at 100 V for 70–90 min. The Trans-Blot^®^ Turbo™ System and Mini Nitrocellulose Transfer Packs (BioRad, Hercules, California, USA) were used. Blots were incubated with primary antibodies (anti-H3 1:2000, anti-acetyl-H3K9 1:1000, anti-HDAC 1 1:1000, and anti-HDAC 2 1:1000; all from Cell Signaling Technology, Danvers, MA, USA) overnight at 4 °C. After washing, blots were incubated with the secondary antibody (anti-rabbit IgG HRP-linked 1:1000, Cell Signaling Technology) for one hour at room temperature. Blots were developed and analyzed using Signal Fire ECL Reagent (Cell Signaling Technologies, Danvers, MA, USA) and a ChemiDoc MP System (BioRad). Protein expression was quantified by the calculation of gray densities using ImageJ and related to those of H3. The protein of interest and reference protein were run and processed on separate gels and membranes

### 2.6. Immunohistochemistry of HDAC 1/2 in BTC Cell Blocks and FFPE Samples

Cells were seeded in 60 mm dishes and allowed to grow overnight. Cell blocks were prepared using a 1:1 mix of citrate plasma and Thromborel S (Siemens Healthcare, Marburg, Germany). Immunohistochemistry for HDAC 1 and 2 was performed on cell blocks of BTC cells and non-tumor MMNK-1 cells and on routinely archived FFPE BTC tissue samples (*n* = 78 archived between 1997 and 2017 at the Institute of Pathology (Paracelsus Medical University, Salzburg, Austria)). FFPE BTC samples comprised *n* = 39 intrahepatic cholangiocarcinomas, *n* = 22 perihilar cholangiocarcinomas, *n* = 7 extrahepatic cholangiocarcinomas, and *n* = 10 gallbladder cancers. In brief, 4 µm sections were mounted on glass slides, deparaffinized with graded alcohols, pretreated with high pH for 64 min using compare cell conditioning 1 (Ventana, Tucson, Arizona USA), and stained with the primary antibodies (listed in Table 1) on a Benchmark Ultra platform (Ventana) with the ultraView Universal DAB Detection Kit (Ventana).

The results of IHC staining were carried out by assessing the extensity (% positive cells) and intensity of IHC staining (0–3) of three different representative microscope fields and expressed semi-quantitatively using the quickscore method by multiplication of the extensity and intensity (yielding values between 0–300) for each field [39]. Immunohistochemical analyses (IHC) of human BTC samples were carried out on anonymized specimens according to the local ethics committee (Reference Numbers. 415-EP/73/37-2011 and ECS 1179/2020).

### 2.7. Cytotoxicity of Combined Romidepsin and Cisplatin Treatment

Cells were seeded in a 96-well transparent microplate and allowed to grow overnight. Cells were either treated with cisplatin alone (10-step 1:2 dilution series starting at 20 µM; dilution series range: 100–0.20 µM) or cisplatin and a sub-lethal concentration of romidepsin (0.5 nM for KKU-055 cells, 0.2 nM for TFK-1 cells). Viability was measured after 24, 48, and 72 h, respectively, using the resazurin assay and a Spark multimode reader (Tecan). IC_50_ values were calculated using four-parameter logistic regression.

### 2.8. Statistics

All data points represent mean values of at least three biological replicates (n ≥ 3 independent experiments; each biological replicate contained—if applicable—an appropriate number of technical replicates) ± standard error of mean (SEM). The distribution of clinicopathological nominal and ordinal data were investigated with one-sample binomial or chi-squared tests. The Student’s t-test and the ANOVA, respectively, were applied after analysis of normal distribution for calculation of significances between control and treated samples (in vitro) as well as for more than two subgroups (in situ). Correlation analysis was performed using a calculation of the Pearson’s correlation coefficient. Cut-off values for HDAC 1 and HDAC 2 scores were determined for the mean IHC scores using the receiver operating characteristic (ROC) calculation and Youden Index analysis for overall survival. Survival curves were generated using the Kaplan Meier method and compared by log rank tests (Mantel–Cox). All calculations were performed using OriginPro 9.1 (OriginLab, Northampton, MA, USA) and SPSS v24 (IBM, Armonk, New York, New York, USA). Statistical results were considered significant (*) or highly significant (**) at *p* < 0.05 and *p* < 0.01, respectively.

## 3. Results

### 3.1. HDACs Are Expressed in BTC Cells

To obtain the first general evidence on the involvement of HDACs in BTC, we measured the mRNA and protein levels of all class I, class IIa, class IIb, and class IV *HDACs* in the in vitro BTC model. Figure 1A shows the expression profiles of the individual *HDACs* (1–10, *HDAC 11* was not detectable at mRNA level; see Appendix A for mRNA expression of each individual *HDAC*). *HDACs 1–8* were expressed in all tested cells. Expression levels of *HDACs 9* and *10* were strongly cell-line-dependent with high expression in the CCC-5 cell line. We were not able to measure mRNA levels for *HDAC9* in TFK-1 cells and for *HDAC 10* in HuCCT-1 cells. In general, *HDACs 1* and *2* showed the highest mRNA expression, whereas *HDACs 4*, *5*, and *6* generally showed relatively low mRNA levels. Interestingly, with the exception of *HDACs 4* and *5*, CCC-5 cells displayed the highest HDAC mRNA levels.

Protein levels of HDACs were measured by immunohistochemistry on cell blocks. As shown in Figure 1B, cells expressed HDAC 1–11 proteins in a cell-line-dependent manner (see Appendix A for protein expression of each individual HDAC). With the exception of HDAC 1 and 2, non-tumor MMNK-1 cells showed relatively low HDAC protein levels compared to BTC cells. Of note, we observed significant differences in expression levels regarding some HDACs, e.g., HDACs 7 and 9 were expressed at mRNA levels, whereas we were not able to detect significant protein levels.

### 3.2. BTC Cells Show Different Sensitivity towards HDAC Inhibitors

To obtain a broader understanding of the consequences of HDAC inhibition on BTC, we used a panel of six established HDACis including two pan-inhibitors (belinostat, vorinostat) and four HDAC class-specific inhibitors (mocetinostat, romidepsin, tubastatin A, LMK-235) covering the classical HDAC classes I, IIa, IIb, and IV (see Table 2). The inhibitors were chosen based on their described specificities and, if possible, on the number of current clinical trials and thus their clinical applicability. Since inhibitors targeting HDAC class IIa used in clinical trials are rather unspecific (e.g., pracinostat), we decided to use the more specific HDAC class IIa inhibitors LMK-235 and tubastatin-A, albeit these inhibitors are not part of clinical studies yet. For HDAC class IIb, no specific inhibitor is currently available.

All used HDACis reduced the viability of BTC cells in a concentration- and cell-line-dependent manner (Figure 2A–F, see Appendix A for statistical analysis). However, we observed significant differences regarding the concentration ranges of the inhibitors, resulting in reduced survival. Cells were most resistant to the HDAC 6 inhibitor tubastatin A, with IC_50_ values ranging from about 20 to about 60 µM (Figure 2E). At the other end of the spectrum, treatment of BTC cells with the class I inhibitor romidepsin led to a significant reduction of viability, resulting in IC_50_ values ranging from about 3 to 15 nM (Figure 2D). In the tested concentration range, no reduction of cell viability was quantifiable for CCC-5 cells following romidepsin treatment. For the remaining inhibitors (pan-inhibitors belinostat and vorinostat and HDAC-specific inhibitors mocetinostat and LMK-235), we measured IC_50_ values in the low-µM range for most cell lines. Figure 2G shows a sensitivity profile of the tested cell lines and HDACis. Based on this profile, CCC-5 cells showed a general resistance toward HDAC inhibition, resulting in relatively high IC_50_ values (belinostat, tubastatin A) or no considerable decline in cell viability (IC_50_ calculation not possible for LMK-235, mocetinostat, or vorinostat). This is especially interesting as CCC-5 cells display high HDAC mRNA levels (see Figure 1A). TFK-1 cells are most sensitive toward HDAC inhibition, reflected by relatively low IC_50_ values for most tested inhibitors. Correlation analysis of the IC_50_ values of the used HDAC inhibitors is shown in Figure 2H. Since we observed high sensitivity of BTC cells toward romidepsin, we additionally evaluated the effect of romidepsin on BTC cell viability using the SRB assay to bolster the results obtained from the resazurin assays. Similar to the resazurin assay, results obtained from the SRB assay show high sensitivity of BTC toward romidepsin treatment with IC_50_ values in the (low) nM range (Appendix A).

### 3.3. BTC Cells Show Heterogeneous HDAC 1/2 Activity

Due to the high efficiency of romidepsinin BTC cells, we decided to further investigate this substance as a potential anti-BTC drug. We used the HDAC-Glo I/II Assay and Screening System (Promega) to evaluate HDAC1/2 activity (as described in Materials and Methods) and Western blot to measure HDAC 1/2 protein levels to analyze whether HDAC 1/2 activity and/or expression correlates with the sensitivity to romidepsin. As illustrated in Figure 3A, BTC cells show a heterogeneous HDAC 1/2 activity, with KKU-055, OCUG-1, and TFK-1 cells showing the highest levels. Similarly, BTC cells show heterogeneous HDAC 1/2 protein expression patterns, especially regarding HDAC 2, where OZ and TFK-1 cells display low levels (Figure 3B,C). 

### 3.4. Romidepsin Causes Apoptosis and Secondary Necrosis in BTC Cells

For subsequent experiments regarding romidepsin, we selected the two BTC cell lines KKU-055 and TFK-1. TFK-1 cells display the highest sensitivity to romidepsin treatment as well as high HDAC 1/2 activity, and KKU-055 cells show the highest HDAC 1/2 activity as well as general high sensitivity to HDAC inhibition.

To determine the mode of cytotoxicity of romidepsin in BTC cells, we performed time-resolved analysis of cell viability. Due to the low IC_50_ values of TFK-1 and KKU-055 cells, we adjusted the range of the dilution series (10 step 1:2 dilution series, 100–0.2 nM). In KKU-055 cells, for concentrations ≥6.3 nM, a clear cytotoxic effect was already observable after 24 h of treatment, which resulted in an even more significant decline of cell viability after 30 and 48 h of romidepsin incubation, respectively (Figure 4A). Treatment of cells with intermediate concentrations of romidepsin (3.1, 1.6 nM) resulted in a medium cytotoxic effect, whereas concentrations ≤0.8 nM showed no cytotoxicity even after 48 h (Figure 4A, see Appendix A for statistical analysis). Similarly, romidepsin at concentrations ≥3.1 nM strongly reduced cell viability of TFK-1 cells after 48 h, whereas concentrations ≤0.4 nM had no effect (Figure 4B, see Appendix A for statistical analysis). These time-resolved viability data indicate that romidepsin has a direct cytotoxic effect in BTC cells.

To further elucidate the cytotoxic mode of romidepsin in BTC cells, we performed a RealTime-Glo^TM^ Annexin V Apoptosis and Necrosis Assay and measured fluorescence and luminescence every 30 min for 48 h under cell incubator conditions (5% CO_2_, 37 °C, humidified environment). For both cell lines, treatment with 5 nM romidepsin resulted in a strong initial increase in the luminescence signal (indicating phosphatidylserine presence in the outer leaflet of the plasma membrane) and only a delayed increase in the fluorescence signal (indicating an intact plasma membrane), thus unraveling early apoptosis (Figure 4C,D). At later time points (>30 h for KKU-055 cells and >40 h for TFK-1 cells), we observed a strong increase in the fluorescence signal, indicating loss of membrane integrity. Based on these data, we conclude that romidepsin causes apoptosis (followed by secondary necrosis in vitro) in BTC cells.

### 3.5. Romidepsin Reduces HDAC Activity in BTC Cells

We next evaluated the effect of romidepsin regarding its ability to reduce HDAC activity in BTC cells. As romidepsin preferentially causes acetylation of lysine residues on histone H3 and this effect was seen for several tumor cell lines especially for lysine 9 of histone 3 (H3K9Ac) [45], we used H3K9Ac as one readout of the epigenetic effect of romidepsin. As shown in Figure 5A,B, treatment of KKU-055 and TFK-1 cells with 5 nM romidepsin resulted in a significant increase in H3K9Ac after 24 h. In addition, for both cell lines, we observed a significant decline in HDAC activity (Figure 5C) following romidepsin treatment. These data underline the potential of romidepsin as a specific HDACi in BTC cells.

### 3.6. Romidepsin Augments the Toxicity of the Standard Chemotherapeutic Cisplatin in KKU-055 Cells

Since HDACis can augment the efficiency of standard chemotherapeutics in BTC [6,12], we combined romidepsin with cisplatin to evaluate a potential synergistic effect. Interestingly, the toxicity of cisplatin was augmented only in KKU-055 cells by the addition of romidepsin (Figure 6A–D). We found no such effect in the other tested cell lines (including romidepsin-sensitive TFK-1 cells), as illustrated by similar IC_50_ values between cells treated with cisplatin only and cell treated with cisplatin and romidepsin (Figure 6E; Appendix A), which suggests a cell-line-specific effect for KKU-055 cells. To elucidate whether priming of KKU-055 cells with romidepsin also enhances the cytotoxic effect of cisplatin, we pre-treated KKU-055 cells for 48 h with sub-lethal concentrations of romidepsin before cisplatin treatment. As shown in Figure 6F, priming of KKU-055 cells with 0.5 nM romidepsin (significantly) augmented the cytotoxic effect of cisplatin and reduced the cisplatin IC_50_ value.

### 3.7. HDAC 1 and 2 (HDAC 1/2) Are Expressed in BTC Patient Samples

To assess whether HDAC 1/2 are expressed in BTC patient samples and correlate with clinicopathological data, we measured their protein expression in *n* = 78 BTC cases by immunohistochemistry. Table 3 summarizes the patient characteristics: the 78 BTC cases included *n* = 39 intrahepatic (50.0%), *n* = 22 perihilar (28.2%), and *n* = 7 extrahepatic (9.0%) cases of BTC, as well as *n* = 10 GBC cases (12.8%). As shown in Figure 7A,B, HDAC 1 and 2 are heterogeneously expressed in BTC patient samples (see Appendix A for description of ROC analysis). Moreover, we found that in adjacent non-tumor tissue, HDAC 1 and 2 were expressed at low levels and independent of the expression levels in the tumor area, meaning that in tumor samples with high HDAC 1 or 2 expression, the expression of these proteins was considerably lower in the respective adjacent non-tumor counterparts (Figure 7C,D). These findings are comparable to an in silico analysis for HDAC 1 and 2 mRNA expression in BTC tissues compared to non-tumor tissue (http://gepia.cancer-pku.cn [46], accessed on 6 July 2021) as shown in Appendix A. Figure 7E shows representative images of low, medium, and high HDAC 1 and 2 immunohistochemistry as well as adjacent non-tumor tissue (small images), respectively. HDAC 1 expression significantly differed between tumor localizations (Table 3). Moreover, HDAC 1 expression was significantly higher in high-graded tumors (G3/G4) compared to low-graded tumors (G1/2). For HDAC 2, these associations were similar but not statistically significant (Table 3).

As shown in Figure 8A, HDAC 1 has a small beneficial effect on the overall survival of BTC patients, whereas those with high expression of HDAC 2 had a significantly shorter overall survival time (Figure 8B). Cases with low versus high HDAC 1 or HDAC 2 expression were defined by ROC analysis and the Youden Index (Appendix A). The superior discrimination potential of the used Youden Index-based cut-off values over mean values, median values, and absolute two-year overall survival data was confirmed by incremental time-dependent logistic regression analysis with an SPSS macro (Appendix A). Moreover, the discrimination power of our Youden Index-based cut-off values is superior to the discrimination power of the default cut-off values (median HDAC 1/2 mRNA expression) of the in silico analysis (http://gepia.cancer-pku.cn, accessed on 6 July 2021) related to the corresponding data sets (Appendix A). Correlation of HDAC 2 expression with markers of epithelial-to-mesenchymal transition (EMT) and histone methyltransferase G9a was moderate and significant for vimentin (positive correlation) and E-cadherin (negative correlation) when measured at the tumor margins (Figure 8C). Moreover, we found a significant moderate positive correlation between HDAC 1 and 2 expression with protein levels of the histone methyltransferase G9a.

## 4. Discussion

In the present study, we confirmed that members of the HDAC classes I, II, and IV are heterogeneously expressed in BTC cells, as already shown in other in vitro studies: Sakamato et al. showed high HDAC class I levels, especially in chemoresistant BTC cells [27]. Likewise, Sriraksa et al. used three BTC cell lines and found differential HDAC class I and class II expression on mRNA and protein level for HDACs 1–10 [47]. Interestingly, the cell line CCC-5 displayed the highest mRNA levels of most tested HDACs, whereas it only expressed moderate protein levels compared to the other cell lines. This might be explained by additional post-transcriptional regulation of HDAC expression [48]. Furthermore, despite having the highest mRNA of most HDACs, the CCC-5 cell line shows a general resistance toward HDACi, which might indicate a cell-line-specific phenomenon, as we did not find similar results and patterns within our in vitro model. Inhibition of HDACs is an established anti-tumor strategy [12]. In our study, we performed a comprehensive screening of the effect of different HDACis covering the various HDAC classes of BTC cell survival. We found that all used HDACis reduced cell viability of BTC cells in a cell-line-dependent manner, which indicates a general efficiency of HDACi in BTC. Interestingly, we found that treatment with both HDAC pan-inhibitors as well as HDACi specific to individual classes resulted in a comparable reduction in BTC cell viability. In our BTC cell-line model, the pan-inhibitor vorinostat reduced cell viability with IC_50_ values ranging between about 4.0 and 15.0 µM—i.e., in the low-µM range, which is slightly higher than in another study using different BTC cell lines [47]. Other studies further underlined the potential of HDAC inhibition in BTC regarding cell migration and cell viability, although these studies only used pan-inhibitors and not class-specific HDACis and/or only a limited number of BTC cell lines [25,26,28,29]. Our results complement the existing data but also identified the HDAC class I inhibitor romidepsin as a substance that reduces BTC cell viability at low concentrations. Up to now, there have been no data available regarding the effect of romidepsin in BTC. However, studies in other tumor entities also demonstrate a significant anti-tumor effect of romidepsin [17,18,49]. In line with our results, Chang et al. described IC_50_ values in the low-nM range for several tumor in vitro models (breast cancer, lung cancer, and melanoma) [49]. Interestingly, in this study, treatment of tumor cells with romidepsin did not reduce HDAC 1 and HDAC 2 activities (even at relatively high doses of 50 nM [49]), whereas, in our study, we observed a clear decline of HDAC 1/2 activity after romidepsin treatment in BTC cells. Clarke et al. [18] demonstrated a cytotoxic effect of romidepsin after 48 and 72 h of treatment at a low-nM range in MDS (myelodysplastic syndrome) and AML (acute myeloid leukemia) cells. Similar to our own data, Clarke et al. observed a significant reduction of HDAC activity accompanied by an increase in H3K9Ac caused by 5 nM romidepsin. Interestingly, in contrast to our results, Clarke et al. [18] did not observe apoptotic events after 24 h and only moderate levels of apoptosis after 48 and 72 h, respectively, which, again, might indicate tumor-specific mechanisms. Besides its effect as a single drug, we also found that romidepsin augmented the cytotoxicity of cisplatin, which is especially relevant for BTC, as cisplatin is part of the standard chemotherapeutic regimen. However, this effect was only seen in one of the tested cell lines (KKU-055), suggesting a cell-line-dependent rather than general effect. Some studies described a connection between HDAC expression and chemoresistance as well as a synergistic cytotoxicity of combined HDACi and chemotherapeutic treatment in BTC. Sakamoto et al. not only demonstrated that HDAC 1 was more expressed in gemcitabine-resistant BTC cells compared to parental cells but also that treatment of BTC cells with the HDACi vorinostat attenuated gemcitabine resistance [27]. Asgar and coworkers showed that a combination of cisplatin with HDAC pan-inhibitors TSA and SAHA resulted in a synergistic cytotoxic effect and enhanced apoptosis [28]. Likewise, two other studies demonstrated an augmentation of the cytotoxicity of the chemotherapeutic 5-fluorouracil in BTC cells when combined with SAHA and valproic acid, respectively [47,50]. Since both valproic acid and romidepsin inhibit HDAC 1, it will be interesting to investigate whether inhibition of HDAC 1 combined with other chemotherapeutic substances or epigenetic inhibitors might be a potent anti-BTC strategy.

Based on the observed sensitivity of BTC cells toward romidepsin, we evaluated the clinical relevance of HDAC 1 and 2 expression in BTC patient samples. We found that HDAC 1 protein expression was significantly different for clinical BTC cases at different tumor localizations and that HDAC 1 expression significantly increased with tumor grading (similar, but not significant for HDAC 2). Likewise, Morine et al. [20] were able to associate HDAC 1 expression with unfavorable clinical characteristics in BTC. In contrast to Morine et al., we found no correlation between HDAC 1 expression and clinical outcome. Du et al. analyzed HDAC 2 expression in *n* = 136 BTC samples and found significant correlations regarding histological grade, clinical stage, and lymph node metastasis [22]. In our BTC cohort, HDAC 2 expression did not correlate with these clinicopathological data, which might be due to cohort-specific expression characteristics or the lower number of investigated cases. We found a small beneficial effect of HDAC 1 protein expression on the overall survival of BTC patients, which is in contrast to other findings [20], whereas high HDAC 2 expression correlated with shortened survival, as already demonstrated by Du et al. [22]. It is known that different HDACs can have different effects on survival and prognosis. In fact, in one of our recent works, we have shown in pancreatic neuroendocrine tumors that high HDAC 1 expression was linked to longer survival, whereas high HDAC 2 expression correlated with bad prognosis, which mirrors the results of the present study [51]. Therefore, more studies are required to elucidate the negative or positive association between different HDACs and survival in BTC.

We have previously shown that other epigenetic regulators (the histone methyltransferase G9a and the histone-methylating polycomb repressive complexes (PRC) 1 and 2) are also involved in BTC [52,53,54]. There is evidence that HDACs interact directly and indirectly with other histone-modifying complexes. EZH2 is the histone methyltransferase of the PRC2, another epigenetic regulator that is associated with gene silencing (in BTC) [55,56]. Several studies found a direct interaction between HDAC class I and EZH2 [57,58]. Interestingly, Yamaguchi et al. showed that combined inhibition of HDAC class I (via SAHA) and EZH2 (via siRNA) resulted in a synergistic cytotoxic effect as well as in increased protein levels of the tumor suppressors E-Cadherin and p16, accompanied by decreased promoter binding [59]. Shi et al. [60] showed that HDAC 2 directly interacts with G9a via the co-repressor CtBP and that this coordinated interaction between HDAC 2 and G9a resulted in silencing of E-Cadherin. The observed moderate but significant correlation of HDAC 1/2 and G9a in the clinical BTC samples might be preliminary evidence for such a cooperative mechanism of HDAC and G9a in BTC. Therefore, based on the promising results of our study regarding romidepsin as an anti-BTC substance, future studies should also investigate the potential of combined treatment of BTC cells with romidepsin and inhibitors of the G9a, PRC1, and PRC2 as regulators of the epigenetic machinery, respectively.

Although our results contribute important findings to a field with sparse data, our study possesses several limitations. Although the kit used in the current study is more specific to HDAC 2 than HDAC 1 and other HDAC isoforms, we cannot exclude that the activity of other HDACs influences the results of our measurements. Furthermore, our data are based on an in vitro model and retrospective analysis of BTC patient samples. We suggest further prospective in vivo studies to extend the knowledge regarding romidepsin as an anti-BTC substance.

## 5. Conclusions

In summary, we demonstrate in a comprehensive BTC in vitro model that HDACs are differentially expressed in BTC cells and that BTC cells show a heterogeneous sensitivity to different HDAC inhibitors. Specifically, we found that BTC cells are sensitive to the HDAC class I inhibitor romidepsin. Combined with our observation that high HDAC 2 expression correlates with reduced survival, we suggest that romidepsin can be an additional option regarding the treatment of BTC.

## Figures and Tables

**Figure 1 cancers-13-03862-f001:**
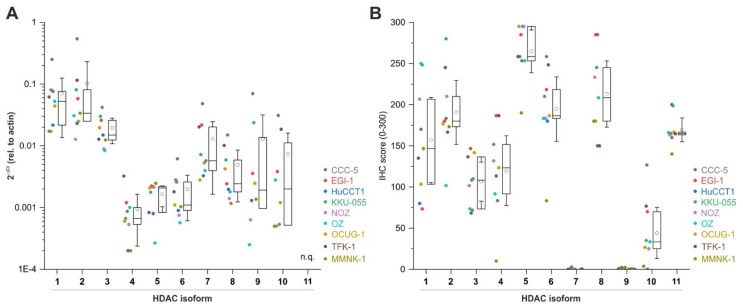
Expression analysis of HDACs in BTC cell lines. (**A**) mRNA levels of HDACs 1–11 in *n* = 8 BTC cell lines and non-tumor cholangiocytes MMNK-1 (HDAC 11 not quantifiable (n.q.) on mRNA level). (**B**) Protein expression of HDACs 1–11 in BTC and non-tumor cholangiocytes MMNK-1 cell blocks by immunohistochemistry (IHC). Data are presented as mean value of *n* = 3 biological replicates. Box plots indicate the 25/75 percentiles; whiskers stand for 95% confidence intervals. Abbreviations: BTC, biliary tract cancer.

**Figure 2 cancers-13-03862-f002:**
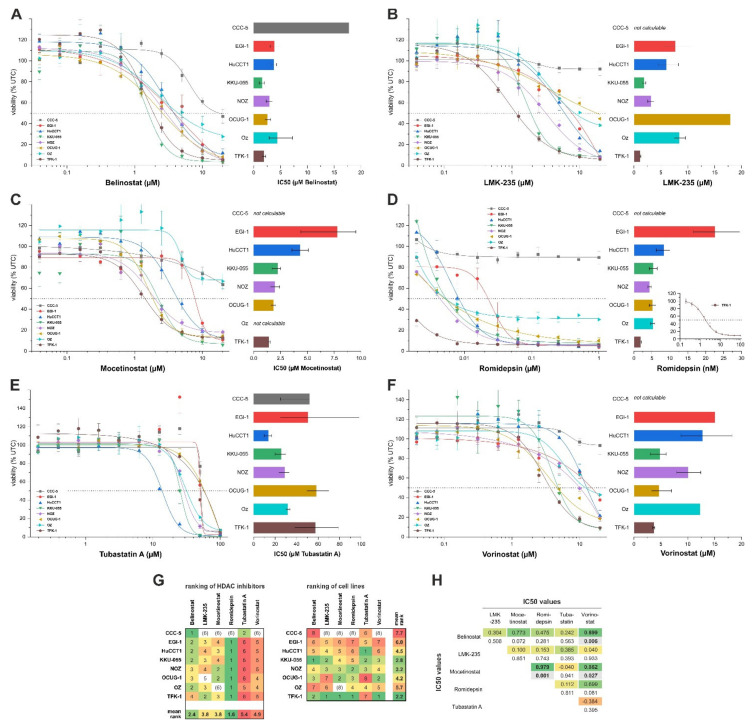
Effect of HDAC inhibitors on viability of BTC cells.(**A**–**F**) BTC cells were incubated for 72 h with pan-inhibitors belinostat and vorinostat, the HDAC I and IV inhibitor mocetinostat, the HDAC I inhibitor romidepsin, and the HDAC class IIa inhibitors LMK-235 and tubastatin-A, respectively. Shown are viability data related to untreated controls (UTC) as well as the respective IC_50_ values for each cell line and each inhibitor based on four-parameter logistic regression (error bars indicate the 95% confidence interval). (**G**) Based on the calculated IC_50_ values, (i) HDAC inhibitors were ranked according to the magnitude of their ability to reduce BTC cell viability (the lower the IC_50_, the lower the ranking of the inhibitor; left panel), and (ii) BTC cells were ranked according to their respective sensitivities (the lower the IC_50_, the lower the ranking of the cell line; right panel). (**H**) Correlation analysis between the IC_50_ values of the used HDAC inhibitors. Green boxes indicate a positive correlation, bold font a significant correlation (*p* < 0.05). Data are presented as mean value ± SEM related to untreated control cells of at least three individual biological replicates. Abbreviations: BTC, biliary tract cancer.

**Figure 3 cancers-13-03862-f003:**
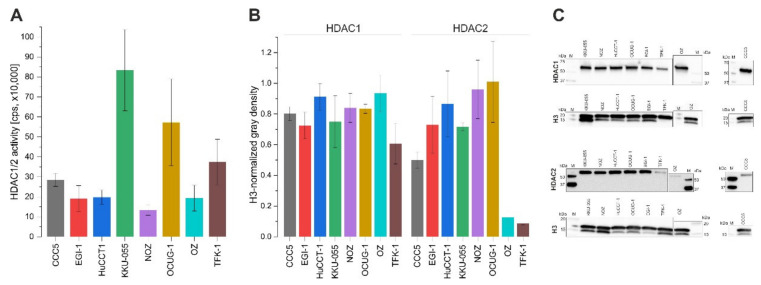
HDAC 1/2 activity and protein expression in BTC cells. (**A**) HDAC activity was measured using an HDAC-Glo I/II Assay and Screening System (Promega). (**B**) Quantification of HDAC 1 and HDAC 2 protein levels in BTC cells. (**C**) Representative Western blot images, cropped. The protein of interest and reference protein were run and processed on separate gels and membranes. For original uncropped blots see Appendix A. Data are presented as mean value ± SEM of *n* = 3 or *n* = 4 (HDAC1 protein expression) individual biological replicates. Abbreviations: BTC, biliary tract cancer.

**Figure 4 cancers-13-03862-f004:**
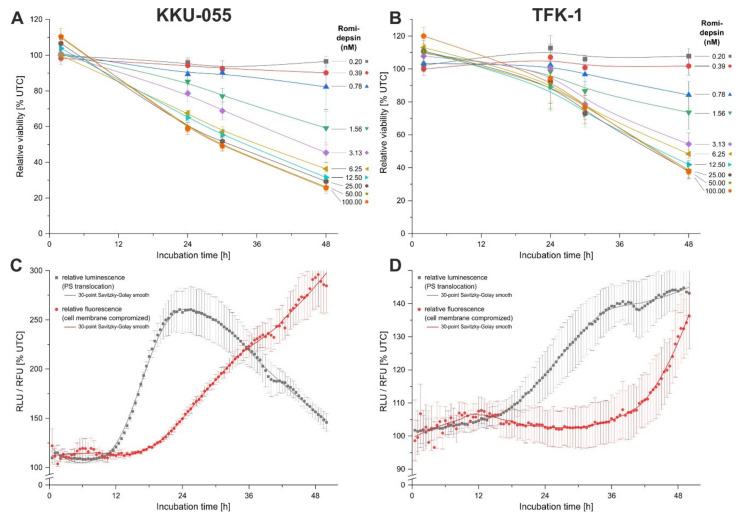
Romidepsin induces apoptosis in BTC cells. (**A**,**B**) Time-resolved analysis of cytotoxicity of romidepsin via resazurin-based endpoint measurement of cell viability in and KKU-055 (**A**) and TFK-1 cells (**B**) indicates direct time- and dose-dependent cytotoxicity. (**C**,**D**) Non-endpoint, continuous measurement of PS translocation and membrane integrity under cell incubator conditions (5% CO_2_, 37 °C, humidity) reveals apoptosis and secondary necrosis in KKU-055 (**C**) and TFK-1 (**D**) cells, respectively. Data are presented as mean value ± SEM related to untreated control cells of *n* = 4 individual biological replicates. Abbreviations: BTC, biliary tract cancer; PS, phosphatidylserine.

**Figure 5 cancers-13-03862-f005:**
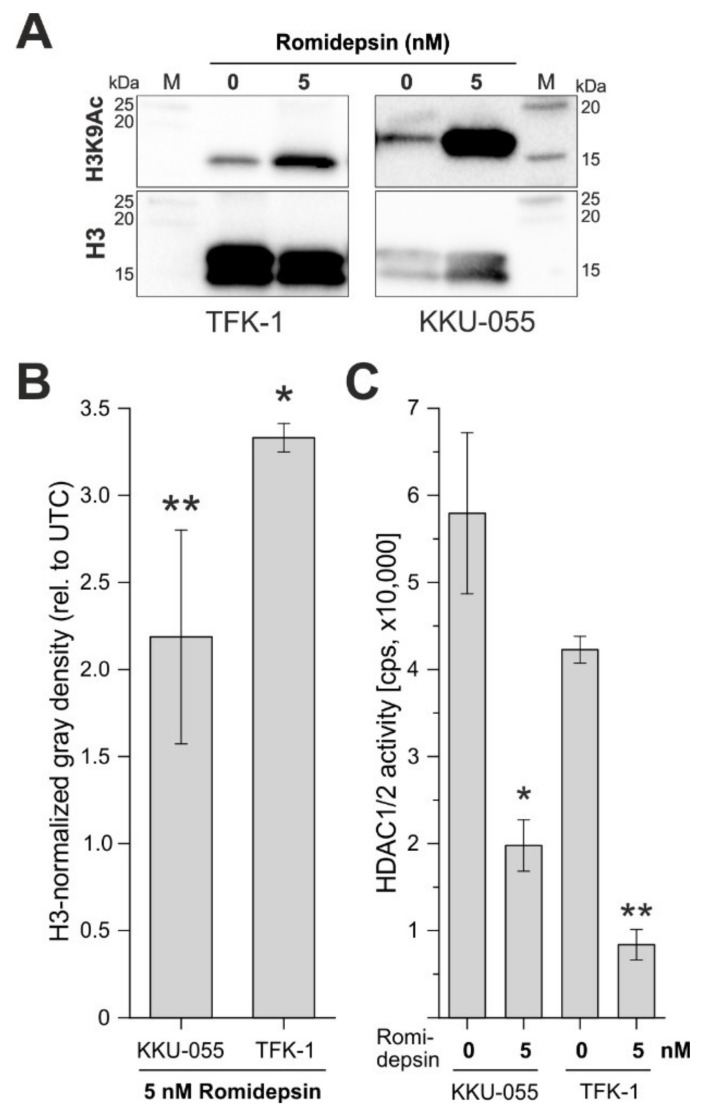
Romidepsin causes H3K9Ac via inhibition of HDAC 1/2 activity. (**A**) Treatment of TFK-1 and KKU-055 cells with romidepsin (5 nM) for 24 h causes an increase in H3K9Ac (representative Western blot images, cropped). (**B**) Quantification of H3K9Ac levels following romidepsin treatment. (**C**) Romidepsin reduces HDAC 1/2 activity in KKU-055 and TFK-1 cells, respectively. HDAC activity was measured using an HDAC-Glo I/II Assay and Screening System (Promega). The protein of interest and reference protein were run and processed on separate gels and membranes. For original uncropped blots see Appendix A. Data are presented as mean value ± SEM of *n* = 3 (Western blot) or *n* = 4 (HDAC activity) individual biological replicates. Asterisks indicate significant (* *p* < 0.05) or highly significant (** *p* < 0.01) differences related to untreated control cells. Abbreviations: cps, counts per second; H3K9Ac, acetylation of histone 3 at lysine 9; M, marker.

**Figure 6 cancers-13-03862-f006:**
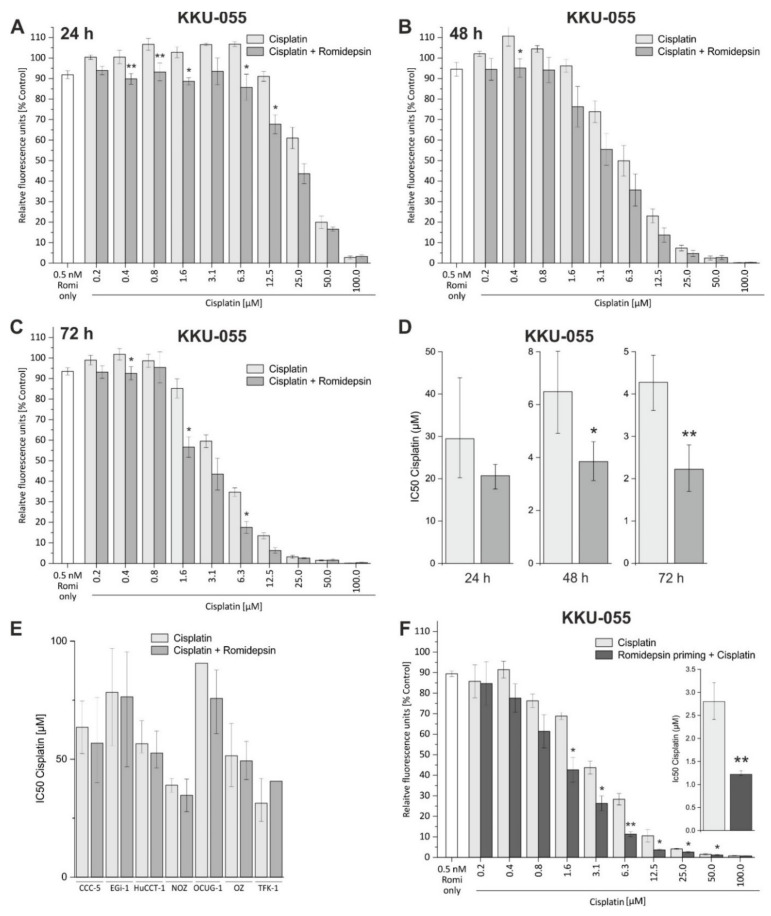
Romidepsin augments the cytotoxic effect of cisplatin in KKU-055 cells. (**A**–**C**) Simultaneous treatment of KKU-055 cells with a dilution series of cisplatin and 0.5 nM romidepsin for 24 h (**A**), 48 h (**B**), and 72 h (**C**), respectively, augments the cytotoxic effect of single cisplatin treatment for specific cisplatin concentrations. The left bar (white) represents the cytotoxic effect of 0.5 nM romidepsin as a single treatment. (**D**) Comparison of IC_50_ values between cisplatin-only and combined romidepsin and cisplatin treatment for KKU-055 cells (error bars indicate the 95% confidence interval). (**E**) IC_50_ values for cisplatin-only and combined simultaneous cisplatin romidepsin treatment for remaining BTC cells after 72 h treatment (error bars indicate the 95% confidence interval) (**F**). Pre-incubation of KKU-055 cells for 48 h with 0.5 nM romidepsin (priming) before the addition of cisplatin (dilution series). The respective IC_50_ values are shown in the insert (error bars indicate the 95% confidence interval). Data are presented as mean value ± SEM of *n* = 3 (KKU-055 cells) or *n* = 4 individual biological replicates. Asterisks indicate significant (* *p* < 0.05) or highly significant (** *p* < 0.01) differences. Abbreviations: BTC, biliary tract cancer.

**Figure 7 cancers-13-03862-f007:**
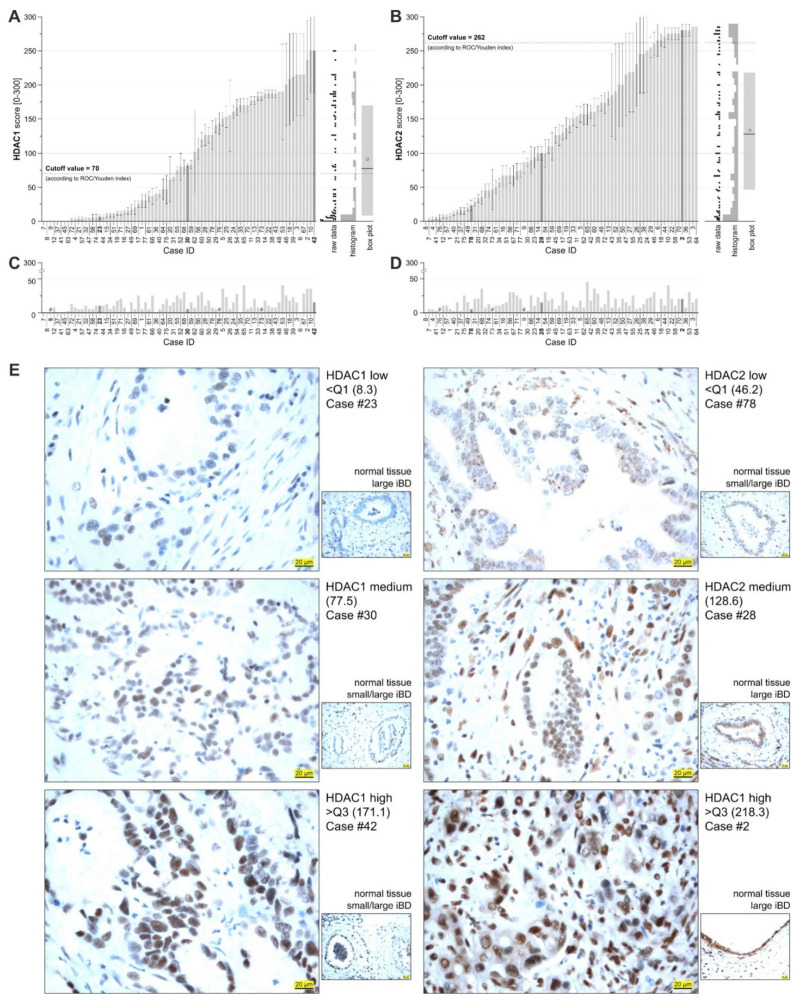
HDAC 1 and 2 are heterogeneously expressed in BTC patient samples. (**A**,**B**) Mean HDAC 1 (**A**) and HDAC 2 (**B**) expression (mean HDAC 1/2 IHC score) of the *n* = 78 BTC cases. Mean HDAC 1/2 IHC scores were calculated based on staining intensity and extensity for three different areas of the specimen. Box plots indicate the 25/75 percentiles. (**C**,**D**) Mean HDAC 1 (**C**) and HDAC 2 (**D**) expression (mean HDAC 1/2 IHC score) of adjacent non-tumor tissue. # = adjacent non-tumor tissue not present on tissue sample. (**E**) Representative IHC images of BTC cases for HDAC 1 (low, medium, high; left panel) and HDAC 2 (low, medium, high; right panel) as well as adjacent non-tumor tissue, respectively. Scale bar (yellow) indicates 20 μm for 40 × magnification. Abbreviations: BTC, biliary tract cancer; IHC, immunohistochemistry.

**Figure 8 cancers-13-03862-f008:**
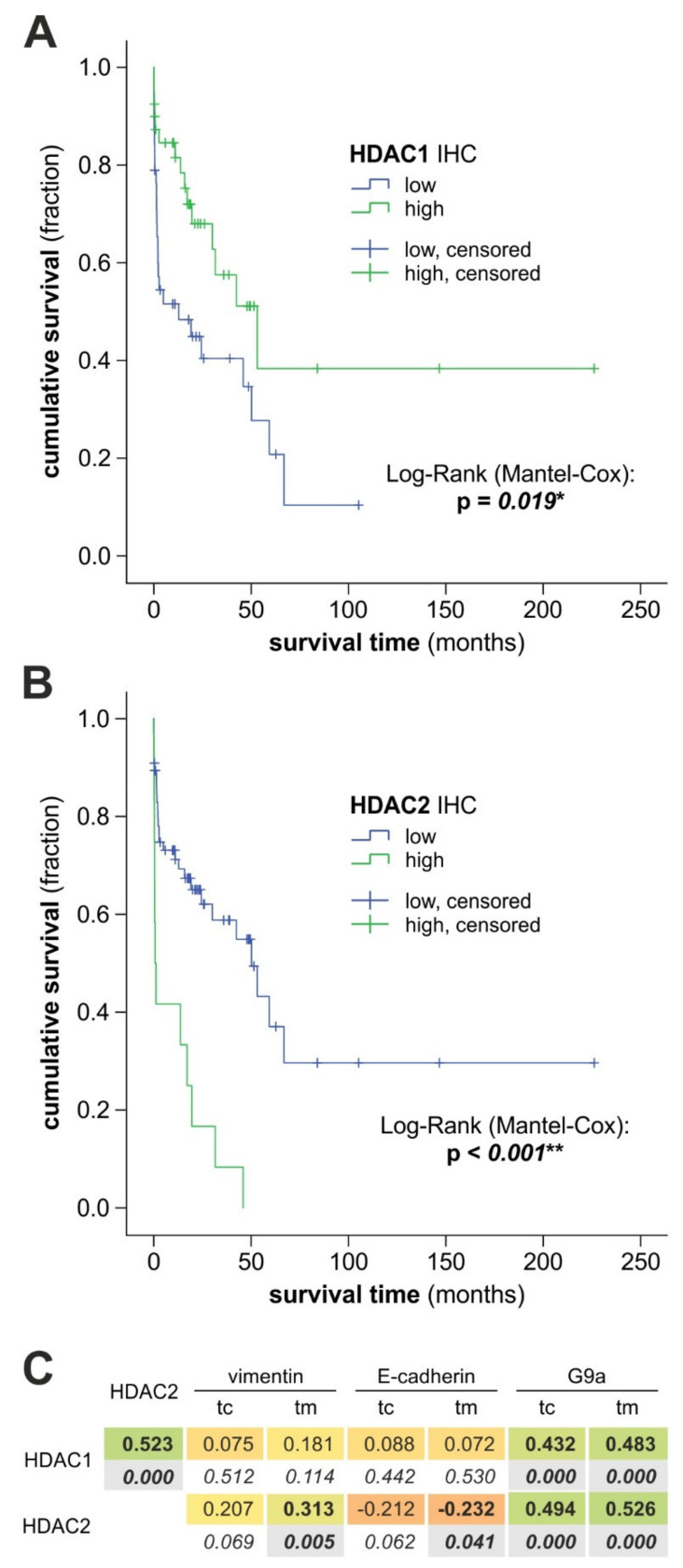
HDAC 1 and 2 expression is related to overall survival in BTC patients. (**A**,**B**) Survival curves comparing cases with high/low HDAC 1 (**A**) or HDAC 2 (**B**) expression. (**C**) Correlation analysis between HDAC 1/2 expression and the level of vimentin, E-cadherin, and G9a. Green boxes indicate a positive correlation, bold font a significant correlation. Abbreviations: BTC, biliary tract cancer; tc, tumor center; tm, tumor margin.

**Table 1 cancers-13-03862-t001:** Immunohistochemistry parameters.

Antibody ^1^	Cat.-No.	Clone	Dilution/Incubation
HDAC 1	ab19845	polyclonal	1:2000/32′
HDAC 2	ab16032	polyclonal	1:250/32′
HDAC 3	ab32369	monoclonal (Y415)	1:100/32′
HDAC 4	ab12172	polyclonal	1:100/32′
HDAC 5	ab55403	polyclonal	1:500/32′
HDAC 6	ab1440	polyclonal	1:500/32′
HDAC 7	ab53101	polyclonal	1:100/32′
HDAC 8	ab195057	polyclonal (Phospho S39)	1:100/32′
HDAC 9	ab59718	polyclonal	1:500/32′
HDAC 10	ab53096	polyclonal	1:200/32′
HDAC 11	ab135492	polyclonal	1:50/32′

^1^ All antibodies (rabbit) were purchased from Abcam (Cambridge, UK). For all stainings, pre-treatment of slides was performed with high pH for 64 min (CC1, compare cell conditioning 1; Ventana, Tucson, AZ, USA).

**Table 2 cancers-13-03862-t002:** HDAC inhibitors.

Inhibito	HDAC Targets	No. Trials ^1^	Ref.
Belinostat	pan-inhibitor	50	[40]
Vorinostat	pan inhibitor	275	[41]
Mocetinostat	class I (HDACs 1, 2, 3); class IV (HDAC 11)	23	[42]
Romidepsin	class I (HDACs 1, 2)	105	[17]
LMK-235	class IIa (HDAC 4, 5)	-	[43]
Tubastatin A	class IIa (HDAC 6)	-	[44]

^1^clinicaltrials.gov (last accessed 25 July 2021).

**Table 3 cancers-13-03862-t003:** Clinicopathological characterization of the BTC patient cohort (*n* = 78) and HDAC 1 and 2 expression by immunohistochemistry (IHC).

Clinico-Pathological Variables		Total		HDAC 1 IHC Score				HDAC 2 IHC Score			
		*n*	%	Mean	Stdev	95% CI	ANOVA	Mean	Stdev	95% CI	ANOVA
Age (years)	<60	14	17.9	94.7	84.1	46.1–143.3	0.878	142.0	116.6	74.7–209.3	0.722
	≥60	64	82.1	90.9	82.6	70.3–111.6		132.0	89.9	109.5–154.4	
Gender	female	36	46.2	95.1	75.6	69.5–120.7	0.733	127.9	91.5	97.0–158.9	0.617
	male	42	53.8	88.6	88.5	61,1–116.2		138.8	97.8	108.3–169.3	
Tumor localization	intrahepatic	39	50.0	83.9	81.9	57.4–110.5	<0.001 **	139.0	103.6	105.4–172.7	0.095
	perihilar	22	28.2	52.1	62.9	24.1–80.0		95.7	73.9	62.9–128.5	
	extrahepatic	7	9.0	125.2	81.7	49.6–200.8		170.2	97.9	79.6–260.8	
	gall bladder	10	12.8	185.1	36.9	158.7–211.5		171.5	74.9	117.8–225.1	
Growth pattern	Mass-forming	34	43.6	83.6	83.5	54.4–112.7	0.427	152.5	100.7	117.3–187.6	0.286
	periductal	41	52.6	94.2	82.6	68.1–120.3		120.9	90.0	92.5–149.3	
	intraductal	3	3.8	147.2	59.0	0.4–293.9		97.7	94.5	n.a.–231.6	
T status (2017)	T1/T2	63 (25/38)	80.8 (32.1/48.7)	90.6	83.2	69.6–111.5	0.823	130.1	93.6	106.5–153.6	0.482
	T3/T4	15 (12/3)	19.2 (15.4/3.8)	96.0	81.5	50.8–141.1		149.3	100.1	93.9–204.7	
N status (2017)	N0	44	56.4	99.7	89.2	72.6–126.8	0.584	145.5	93.1	117.2–173.8	0.409
	N1	26	33.3	84.0	76.0	53.3–114.7		123.2	99.7	82.9–163.5	
	N2	8	10.3	72.0	63.4	19.0–125.1		103.7	84.2	33.3–174.1	
M status (2017)	M0	68	87.2	95.2	81.8	75.4–115–0	0.324	135.0	94.2	112.2–157.8	0.764
	M1	10	12.8	67.5	86.1	5.8–129.1		125.3	101.1	52.9–197.7	
UICC (2017)	I	19	24.4	111.5	91.1	67.5–155.4	0.284	153.9	90.1	110.5–197.4	0.306
	II	21	26.9	106.7	85.9	67.6–145–8		153.4	97.1	109.2–197.6	
	III	24	30.8	77.5	69.5	48.1–106.8		111.8	92.8	72.6–151.0	
	IV	14	17.9	66.4	82.2	18.9–113.9		114.5	97.0	58.5–170.5	
Tumor grading	G1/G2	49 (4/45)	62.8 (5.1/57.7)	68.7	74.0	47.5–90.0	0.001 **	119.2	87.8	93.9–144.4	0.076
	G3/G4	29 (28/1)	37.2 ((35.9/1.3)	130.3	82.4	98.9–161.7		158.5	101.6	119.8–197.1	
Survival	No = dead	38	48.7	106.6	77.0	81.3–131.9	0.119	125.0	78.7	99.1–150.9	0.426
	Yes = alive	40	51.3	77.4	85.7	50.0–104–8		142.1	107.7	107.7–176.6	

** indicate highly significant (*p* < 0.01) results.

## Data Availability

The data presented in this study are available in the article.

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
