# Peer review of "HDAC Screening Identifies the HDAC Class I Inhibitor Romidepsin as a Promising Epigenetic Drug for Biliary Tract Cancer"

_cancers, 2021, doi:10.3390/cancers13153862_

Round 1

Reviewer 1 Report

I appreciate the authors put much effort to fix the issues that were mentioned in the original review. The corrections implemented by the Authors improve the quality of the manuscript, and I recommend this manuscript to publish in Cancers.

Reviewer 2 Report

Thank you for addressing my comments. My only remaining comment is that I don't think it is necessary to show Supplemental figure 10. You can just put the text as is without the figure.

This manuscript is a resubmission of an earlier submission. The following is a list of the peer review reports and author responses from that submission.

Round 1

Reviewer 1 Report

The authors were responsive to some of my concerns. However, a few additional items need to be addressed.

  1. The response to the critique “Table 3 – “statistics” column appears to be meaningless. What is it comparing? It should be removed.” gives further evidence that this column is unnecessary. It would be testing if the categories have equal proportions. It does not makes sense to do this test unless the authors are hypothesizing that 50% of subjects are < 60 years old and >=60 years old, and 50% of subjects are female and 50% are male, and 50% are T status T1/T2 and 50% are T3/4. This column should be removed as meaningless.
  1. Table 3. P-values are never 0.000, they should be reported as <0.001.
  1. Sample sizes for the various experiments should be added to the figure legends. They now say at least 3, but you should give the actual n for each.
  1. The ROC curves don’t seem to be used appropriately. Survival data by definition is censored. I suppose you could define survival at a particular time point, like dead or alive at 2 years. That would contain full information that could be used in a logistic regression model. Even if they were made appropriately, they don’t look very good. Based on the ROC curves the ability to discrimination survival status is poor. A limitation of this approach is that the same data was used to determine a cutoff that was then used on the same data for analysis and hypothesis testing. A rigorous analysis would be to determine the cutoff on an independent dataset from the data it is testing it on.
  1. There should be a section describing the limitations of the study in the discussion.

Reviewer 2 Report

The authors have made some changes in revision. However, the revision did not provide additional experiments to improve the significance of the paper. My comments #7 and #8 were not answered. These experiments are fundamental to support authors’s claim. In addition, the explanation on Figure 3 is not satisfied. The authors has measured overall deacetylase activity instead of HADC1/2 activity. An affinity purification of HDAC1/2 is required for measuring HDAC1/2 activity.

Reviewer 3 Report

I appreciate the authors put much effort to fix the issues that were mentioned in the original review. The corrections implemented by the Authors improve the quality of the manuscript. However, there is still some weakness that should be fixed before the manuscript could be recommended for publication. Please find below specific comments:

1). The Authors statement within the 3.6 section that "romidepsin augments the toxicity of the standard chemotherapeutic cisplatin" is an overinterpretation. The Authors performed additional experiments and tested the remaining cell lines regarding a possible potentiation of the cytotoxic effect of cisplatin and did not observe such romidepsin effect. Thus, we cannot talk about the effect of sensitization of cells to cisplatin by romidepsin since such a relationship was observed only in 1 out of 8 tested lines. Rather, it is correlated with the individuality of a given cell line. Therefore, the Authors should show the calculated IC50 values (within Figure 6) for the remaining cell lines and rewrite the subsection so that it does not mislead the readers.

2). There are missing statistics for Figure 2, 3, 4 and partially for Figure 6, so there is no information on whether the difference is statistically significant.

3). The Authors stated that they used separate gels and membranes for proteins of interest and beta-actin as a loading control. Many WB specialists consider such an analysis of the protein level to be an essential error because it prevents proper analysis. The loading control should be analyzed on the same gel as it determines the potential variation in loading the sample onto the well. If the authors analyzed the loading control on another gel, it should be clearly stated both in the description of the method and the figure's description.

4) Also Authors stated that they “performed the dilution series with DMSO equivalent to the dilution series using the different HDACis. Based on the stocks solutions of the HDACis, the highest DMSO concentration used was 0.1% (1:1000)”. This is an error that may affect the obtained results. Despite using a series of HDACi dilutions, the solvent concentration (DMSO) should remain constant over the entire HDACi concentration range used. Otherwise, the observed effect may be influenced by increasing solvent concentration. On the other hand, at the lowest concentration of the solvent, we can exceed the solubility threshold of the substance and cause it to precipitate in solution and be unavailable to cells.